# Validity and Reliability of the Satel 40 Hz Stabilometric Force Platform for Measuring Quiet Stance and Dynamic Standing Balance in Healthy Subjects

**DOI:** 10.3390/ijerph17217733

**Published:** 2020-10-22

**Authors:** Pere Ramón Rodríguez-Rubio, Caritat Bagur-Calafat, Carlos López-de-Celis, Elena Bueno-Gracía, Rosa Cabanas-Valdés, Ernesto Herrera-Pedroviejo, Montserrat Girabent-Farrés

**Affiliations:** 1Department of Physiotherapy, Faculty of Medicine and Health Sciences, Universitat Internacional de Catalunya, UIC Barcelona, Sant Cugat del Vallès, 08195 Barcelona, Spain; prodriguez@uic.es (P.R.R.-R.); cbagur@uic.es (C.B.-C.); carlesldc@uic.es (C.L.-d.-C.); eherrera@uic.es (E.H.-P.); 2Fundació Institut Universitari per a la recerca a l’Atenció Primària de Salut Jordi Gol i Gurina (IDIAPJGol), 08007 Barcelona, Spain; 3Department of Physiatrist and Nursery, Faculty of Heath Sciences, University of Zaragoza, 50009 Zaragoza, Spain; ebueno@unizar.es; 4Department of Physiotherapy, Blanquerna School of Health Sciences, Ramon Llull University, 08025 Barcelona, Spain; 5Department of Physiotherapy, School of health Sciencies, Tecnocampus-pompeu Fabra University, Mataró, 08302 Barcelona, Spain; girabent@gmail.com

**Keywords:** validity and reliability, postural control, posture, stabilometry, posturography

## Abstract

Background: A force platform must have validity and reliability for optimal use. The objective of this study was to analyze the validity and the reliability of the Satel 40 Hz stabilometric force platform. Methods: A study of instrumental validity and reliability, involving a cross-sectional correlational and comparative analysis was performed. To determine the validity, four certified weights located on three axes were used and the ability of the stabilometric force platform to detect changes in the position of the different axes was observed. A test–retest was performed to analyze the reliability. Forty-two symptom-free volunteers participated in the study. Assessments were taken in a standing static position and in a dynamic position, with the eyes open and closed. Three measurements were taken and the intra-class correlation coefficient (ICC) was calculated. Results: The validity increased as the weight increased for all the variables measured in the stabilometric parameters (*p* < 0.05). The reliability was shown to be good to excellent for the two visual conditions. The positional variables obtained a higher ICC. The variable with the best ICC was the Y mean in OE (ICC 0.874 and a *p* < 0.001). All the values showed an increase in a dynamic situation. Conclusion: The findings support the reliability and validity of the Satel 40 Hz stabilometric force platform. The platform could be recommended to evaluate static and dynamic standing balance in healthy adult individuals. Guidelines for treatment and the level of quality of stabilometry could be obtained from its use.

## 1. Introduction

Postural sway during quiet standing is a reflection of the interplay between the destabilising forces acting on the body, such as gravity and the external environment, and the reaction of the postural control system to prevent the loss of balance [1]. Postural control is the ability to maintain equilibrium in a gravitational field in order to maintain the center of body mass over its base of support or to return it there. When unsupported, standing humans are in unstable balance or equilibrium, because the force of gravity must continually be countered by muscular energy [2].

According to the specialized literature, the force platform is effectively the most used instrument to evaluate postural stability [3,4,5]. It is a device that is sensitive to the forces applied by the individual to the ground. The center of pressure (COP) and its trajectory (postural sway) provide information about the postural response, and classical methods are usually referred to as stabilometry [6]. This is the objective study of body sway during quiet standing, and its stance in the absence of any external disturbances or voluntary movements. Visual inputs during this measurement may be reduced by closing the eyes.

Technical performance parameters for stabilometric assessment instruments must be based on the COP Sway Signal measurement [7]. The COP Sway Signal consists of the x-y time plot of the COP during the test. The x-axis is the horizontal trace of the latero-lateral plane aimed towards the right side of the patient. The y-axis is the horizontal trace of the antero-posterior plane aimed in front of the patient. 

Stabilometry collects information indicating the steady-state functioning of the postural control system, and its success in stabilizing the body against gravity, by examining the properties of procedures directly or indirectly related with postural sway [8]. It is important in both clinical practice and research, as it permits better diagnosis and treatment [9,10,11,12,13]. It enables a more objective assessment as it provides the results immediately after a working session, with better monitoring of the COP and its evolution. It also complements clinical examinations of static balance, such as the Fukuda [14,15] and the Romberg test [16]. 

Stabilometry is increasingly widely used, and has been successfully used in patients with social anxiety disorders [17], in individuals with Parkinson’s disease [18], and to assess the effects of drugs on postural control, by recording the area that these subjects need to maintain balance [19]. 

The Satel 40 Hz stabilometric force platform is a portable platform used in clinical and research settings for its versatility and ease of use [20]. There are several studies that used this platform but the validity and reliability of the Satel 40 Hz stabilometric force platform has not been studied [21,22,23,24,25,26,27,28,29,30,31,32]. This study therefore aims to validate the Satel 40 Hz stabilometric force platform and its reliability by measuring quiet stance and dynamic standing balance.

## 2. Materials and Methods

### 2.1. Study Design

This study of instrumental validity and reliability, involving a cross-sectional correlational and comparative analysis, was conducted at the Physiotherapy Department of the International University of Catalonia. A convenience, consecutive, and non-probability sampling method was used in the study.

### 2.2. Parameters

The following stabilometric parameters were considered for each measurement, using the locations of the COP. Ellipse surface area that the subject needs to maintain balance (S); Mena displacement in the x-axis (Xm) and y-axis (Ym); distance travelled made by COP (D), distance travelled of COP on the x-axis (DX) and y-axis (DY); total amplitude on the x-axis (X-amplitude) and y-axis (Y-amplitude); maximum value on the x-axis (X-maximum) and y-axis (Y-maximum); minimum value on the x-axis (X-minimum) and y-axis (Y-minimum).

The study consisted of two different phases: in phase I the instrument was validated to determine the change in the position of a constant weight on the platform. In phase II, the purpose was to know the reliability of the measurements on people where there is oscillation in their center of mass.

Phase I: The validity of the Satel 40 Hz stabilometric force platform (model PF2002; SATEL SARL, 6 rue du Limousin—31,700 Blagnac; France) was calculated using the v. 33.58C software. The voltage is 230 V, 50/60 Hz, 2 A with USB cable output. It is classified as a class I-type B device. It uses three sensors with a sensitivity per sensor of 2.0 mV/V ± 0.1. The dimensions of this platform are 48 × 48 × 6.5 cm, and it weighs 12 kg. The platform can measure weights between 0 and 100 kg. It has a maximum capacity of 100 kg for each captor, with a sensitivity of 0.0017%. The platform uses three SP4 mark HBM or Scaime captors. The electrical signal goes to an administration card and directly to the platform software, which is version 33.58C for measuring posture-kinetic activities.

Graph paper was attached to the platform, with marks establishing the origin of coordinates (0,0), a coordinate x-axis marked at 3 cm (3,0) and another y-axis marked at 3 cm (0,3) (Figure 1E). The origin of the coordinates was at the bottom left corner of the platform. Weights were used to eliminate intrasubject biological variability, as specified by the *Laboratoire National de Métrologie et d’Essai* [33]. This means that the differences can be associated to the displacement, rather than other factors when recording the stabilometric parameters at the different coordinates on the plane. In specific terms, these measurements were carried out using a weight of 1.990 kg, a weight of 15.149 kg, a weight of 30.271 kg, and a weight of 45.405 kg (Figure 1A–D). 

In order to determine whether the Satel 40 Hz stabilometric force platform was able to detect the changes in the COP on both the x and y axes, and to ascertain whether its variability declines as the weight increases, measurements at the origin (0,0) and with a displacement of 3 cm on the x (abscissa) axis (3,0) were taken in entirely the same way for the y (ordinate) axis (Figure 1F–H). Measurements were repeated 10 times with each weight for each position. 

Phase II: Reliability of quiet stance and dynamic standing balance. For the phase II of the study, a group of healthy university students aged between 18 and 25 years old were recruited. The study was undertaken according to the Helsinki Declaration, and ethical approval was obtained from the Research Ethics Board of the International University of Catalonia before beginning this study. The participation of the subjects was voluntary, and all of them signed an informed consent form before the study. The sample size was calculated establishing an error of α = 0.05 and β = 0.20, a maximum intra-observer disagreement percentage of 5% and a 95% confidence interval amplitude of 0.10. 

All the individuals were active in their daily life, and none had any diagnosed musculoskeletal or neurological disease at the time of the study. Subjects undergoing pharmacological treatment other than vitamin supplements were excluded, as were subjects with otitis or vestibular system effects during the 3 months prior to the study, individuals undergoing any dental treatment, and pregnant women.

The protocol established by the *Association Française de Posturologie* [34] was followed while performing the various stabilometric measurements. The balance measurements of each subject were carried out between 9 a.m. and 1 p.m. Subjects had not engaged in intense physical activity beforehand, and they had rested for between seven and eight hours the previous night. The measurements were always taken by the same evaluator who was familiar with the instrument and software, in order to avoid procedural discrepancies between the evaluators. Measurements were performed in a 2.5 for 5 m soundproofed room, and absolute silence was maintained in the room during the stabilometric measurement. All the measurements were performed with both open eyes (OE) and closed eyes (CE), while standing and barefoot without socks. A foot positioning template was used to position the feet at an angle of 30°, with the heels 2 cm apart. The subjects were placed in front of a white wall with a red plumb line 90 cm away from the platform. A plumb line is a weight suspended from a string used as a vertical reference line to ensure a structure is centered. As the plum line always finds the vertical axis pointing to the center of gravity, it ensures everything is right, justified, and centered. They were asked to keep their arms by the side of their body, and relax as much as possible, without clenching their jaw (Figure 2A,B). Measurements were repeated if the patient coughed, sneezed, yawned, turned his/her head, or performed maximum inhalation. For the static registration, subjects were placed on the platform without any unbalance. For the dynamic registration, a methacrylate plate which produced anteroposterior and lateral imbalance was used. 

A first static measurement was performed for each subject with OE for 51.2 s [6], followed by a second measurement with the same procedure, without the subject getting off the platform with CE. In accordance with Normes 85 [34], the measurements were repeated after a clearing period of 5–15 min, in which the subject was allowed to sit down or engage in low intensity activity. Static record time of 51.2 s was determined by the platform’s ability to collect 40 data per second with a 24-bit analogue-to-digital conversion card (2048 data captured per minute). The exposure time in dynamic was reduced by half to avoid risk of falling of patients with CE.

The results obtained in the dynamic measurements were generated by imbalances with the methacrylate plate in the anteroposterior plane and in the lateral plane. Dynamic measurements were subsequently performed for 25.6 s, on the anteroposterior plane with OE, then with CE, and then the dynamic procedure for the lateral plane was repeated with OE and CE. The measurements were repeated for the second time, with the same clearing period and procedure. 

### 2.3. Data Analysis

Descriptive statistics were calculated for all the variables and expressed as the mean and standard deviation (SD). Normality was analyzed with the Shapiro–Wilk test. To check for differences between values of the same variable in the different positions, Student’s *t*-test or the Wilcoxon test were used, depending on the normality of the data. Percentage increases for all the stabilometric variables were calculated between the central position (0,0) and the positions (0,3) and (3,0). 

The Pearson or Spearman correlation coefficient was calculated to determine the force and direction between the variable measurements. 

The intraclass correlation coefficient (ICC) with a two-way mixed model and absolute agreement type at 95% confidence interval (CI) was calculated to determine the absolute reliability [35]. The ICCs were interpreted according to Portney and Watkins [36] and included 0.00 between 0.25: little to no relationship, 0.25 between 0.50: fair degree of relationship, 0.50 between 0.75: moderate to good relationship, and 0.75 between 0.9: good, and values greater than 0.90 indicate excellent reliability. All the statistical tests were performed with a significance level of α = 0.05. The SPSS 21.0 statistical analysis software package was used.

### 2.4. Ethics Approval and Consent to Participate

The study protocol was approved by the Human Research Ethics Committee of the International University of Catalonia (UIC) with number: FIS-2012-03. All individuals provided informed consent before enrolment. The personal data of them were kept confidential and the data were shared anonymously upon request from the principal researcher.

## 3. Results

### 3.1. Validity

For the S variables, there were mostly statistically significant differences (*p* < 0.05) between the geometric mean positions of x and y. This was not the case for S when the weight of 30.271 kg was displaced 3 cm on the y-axis (0,3) or for the variable Xm with the weight of 1.990 kg and the same displacement.

As regards Xm, the platform was able to detect the object’s change of position from the origin of coordinates to the coordinates (3,0) which increased by 244.98% with the weight of 1.990 kg, and by 362.02% if the weight is 45.405 kg (Table 1). For the Ym variable, there was an increase in the measurement if the weight at the origin was displaced to the coordinates (0,3). However, this did not happen when the movement was on a perpendicular y-axis towards the coordinates (3,0). S increased when the object moved on the x-axis, while the opposite was true when the object moved on the y-axis, except when the weight used was 45.405 kg. In this case it increased on both axes, although the increase on the x-axis was 60% greater.

The distance on the x-axis and the y-axis with the four weights, compiled at the source of the coordinates (0,0) versus the same measurement at position (3,0) and (0,3), had statistically significant differences (*p* < 0.05) except in five cases. The cases for which no statistical significance was obtained were the variables for the distance of the x-axis for the weights of 15.149 and 30.271 kg if the object was moved 3 cm on the x-axis, and the variable distance on the y-axis for the weight of 1.990 kg if the object was moved to the coordinates (3,0) and the weights of 15.149 and 30.271 kg if they were moved to the coordinates (0,3).

The results of the stabilometric parameters in X amplitude, Stabilometry maximum X and minimum X are statistically significant (*p* < 0.05) if the four weights were moved 3 cm on both the x-axis and on the y-axis, except for the Stabilometry maximum and minimum X and an amplitude on Y for the weight of 1.990 kg (Table 2). The dispersion was always greater in these measurements when the weight is 1.990 kg. For Stabilometry amplitude X, the variance of the measurements was 0 for the weights of 15.149, 30.271, and 45.405 kg and for the three positions, except for the measurement with the weight of 45.405 kg in position (0,3). 

When the weight was moved 3 cm on both axes, the Stabilometry x (3,0) values recorded when the weight was 1.990 kg were always less than the values recorded with the other three weights, regardless of the position of the weight. A segment with a positive gradient was observed for the variable Stabilometry xin amplitude, when the weight had been moved on both the x-axis and y-axis. This gradient was more pronounced on the y-axis, indicating a more pronounced increase in Stabilometry if a vertical displacement occurred.

For the variables Stabilometry in Y-amplitude, Stabilometry in maximum and minimum Y, the results were in all cases statistically significant (*p* < 0.05), whether the objects moved 3 cm on the x-axis or if they moved 3 cm on the y-axis for any of the weights moved, except only for the situation of the variable Stabilometry in y-amplitude, with the weight of 1.990 kg if the object was moved 3 cm to the coordinate (0,3).

A high degree of variability was obtained for Stabilometry in Y in the three variables when the weight used was 1.990 kg compared to the other three weights in the same position. Likewise, for Stabilometry in x, the variation was 0 for the weights of 15.149, 30.271, and 45.405 kg and for the three positions, except for the measurement with the weight of 45.405 kg in position (0,3). 

For Stabilometry in Y in amplitude, the values recorded after moving the weight 3 cm on the *x-axis* were higher than the values at the origin (0,0) and higher than the values recorded after moving the weight 3 cm on the y-axis. The maximum y Stabilometry and the minimum Y Stabilometry always had higher values in the measurements taken when the weight was moved on either axis than in the measurements at the origin of the coordinates (0,0).

### 3.2. Reliability of Stabilometry of Posturographic Variables by Correlation and in Intra-Class Correlation Coefficient (ICC)

Forty-two individuals were required to carry out the reliability analysis of the force platform and a group of 42 healthy individuals were recruited (57.24% were women and 42.76% men, and the mean age was 20.98% ± 1.83). The mean height was 1.67 ± 0.10 m, the mean weight was 63 ± 11.7 kg, and the mean body mass index was 22.46 ± 2.81 kg/m^2^. The mean distance of their right foot was 25.5 ± 1.88 cm.

### 3.3. Static Measurement

The correlation coefficient was higher (0.874 with OE and 0.864 with CE) in the values of Ym in the two visual conditions, indicating good reliability. For the ICC, the variables of Xm, S, and DX also had good reliability with CE, while the reliability was moderate with OE, with a wide range in the confidence interval. The ICC was moderate under both visual conditions for the D and DY parameters. The reliability was poor for the distance in y. The correlation was slightly lower in the DY variable than in the DX variable (Table 3).

The results of the stabilometric variables showed a moderate ICC, with maximum and minimum X Stabilometry and amplitude in the two visual conditions. There was good reliability in minimum and maximum Stabilometry in Y with OE and CE and in Stabilometry in amplitude with CE. In the parameter Stabilometry in amplitude with OE, the reliability was poor, and it is not the only value for which there was no statistical significance (Table 3).

### 3.4. Dynamic Measurement

All the values showed an increase in the means and standard deviations of all the variables in a dynamic situation compared to a static situation. Increasing imbalance also increased the difficulty, as seen in the average of the sagittal imbalance in CE, with a value of 1331.16 and 239.64 mm of S in CE in the static situation.

The ICC for all the values had a reliability between moderate and good in the two imbalances, except for the ICC in the variable D in SI with EO and lateral imbalance with EC, which was excellent and in the Stabilometry in Xm of LI with EO, which was poor. The values of all the variables were statistically significant, except for Stabilometry in Xm (Table 4).

## 4. Discussion

The objective of this study was to validate the Satel 40 Hz stabilometric force platform and to know its reliability on measuring the static and dynamic balance. This platform seems to be valid for detecting changes in the position of the center of pressure (CP) increasing its validity as the weight to be evaluated increases. It seems to be reliable for both static and dynamic measurements. 

The Satel 40 Hz stabilometric force platform has good to excellent reliability for all values for stabilometric measurements for static and dynamic balance. It has excellent reliability for dynamic distance and sagittal imbalance with the eyes open (EO), and dynamic distance by lateral imbalance with the eyes closed (EC).

### 4.1. Validity

The various parameters for the validity of the Satel 40 Hz stabilometric force platform were evaluated, and its ability to detect changes in position from 15 kg, using different weights. Weights were used to eliminate changes caused by the subjects’ biological variability, and to be able to state that if any changes were detected, these were due to the movement of the load on the coordinate axes, rather than because of any other possible confounding variables related to the study subjects. 

There was an important difference between the weight of 1.990 kg and the weight of 15.149 kg for all the variables studied, which improved the accuracy of the measurements. The precision of the measurement therefore cannot be guaranteed for weights of less than 15 kg. The platform was designed for measurements of subjects who can remain standing. However, this means it was less reliable for measurements of subjects weighing less than 15 kg, which based on the percentiles of the Spanish population, would be children under 4 years of age [37].

In most cases, when the same object had moved 3 cm on both the x-axis and on the y-axis, the platform was able to detect changes in its position. These were statistically significant, meaning that the platform can be considered valid for measuring the displacement of the COP, especially for weights of over 15.149 kg.

Another important aspect was that values such as the distance on y, with a weight of 45.405 kg, and the distance on x for both the weight of 15.149 and 30.271 kg were not statistically significant due to electrical noise [38]. We believe that this may have interfered with the results, since the concept of electrical noise was innate in all electronic devices. This noise occurs intrinsically, when the signal recorded by the platform’s sensors was processed, and was assumed to be as indicated by the manufacturer, i.e., an error of 2.0 mV/V ± 0.1. The other electrical noise with an effect was the noise extrinsic to the platform. This noise may also have had an effect since it was the noise generated at a point in the system or in the electrical network as a consequence of this connection, as a result of natural systems (climatic changes, vibrations in the building, etc.). The power grid in Spain has a frequency of 50 Hz, and we believe this could be the main source of interference [39]. The validity increases as the weight increases for all the variables measured, and this trend was observed in the stabilometric parameters with the weights used in this study. 

### 4.2. Reliability of Stabilometry

As for the reliability of the stabilometric measurements, one of the major issues was the moderate to good reliability with CE compared to the moderate reliability with OE, as in the case of the variable Xm in a static situation, where this was more pronounced. This situation, in which measurements with CE were more reliable, could basically be explained by two reasons. 

Firstly, it could be due to the fact that with OE, the participant had to spend more time concentrating on the visual point located at a distance of 90 cm. This was in contrast to the study by Kapteyn (1983) [8], who recommend a distance of 3 m to increase the peripheral vision. This would lead to the subject having the difficulty mentioned in many cases concerning maintaining the vertical visual reference located in front of the participant during the measurement. This vertical visual reference was smaller, and its loss may force the person to adjust their center of pressure position. This relationship between vision and postural control had been clearly demonstrated [40,41] by many authors. There was also a possible loss of attention, despite the fact that the examination room was isolated from any noise [42,43] and a constant readjustment of the nervous system by the facilitating and inhibiting systems in order to constantly process the same sensory information that reaches the cortical and subcortical areas for the maintenance of posture. Subjects with an early defect in binocular function, occlusion in one eye, or even the fixation of concentration in only one eye have an imbalance in the information from the entrance to the vestibular system. This creates a latent nystagmus, which sometimes can only be seen with a magnifying glass and is known as dissociated deviation [44,45,46]. Some authors argue that there is a relationship between the mental calculation process and improved postural stability, as maintaining an active mental process enhances the attention system, thereby reducing oscillations in the measurements [34,40,41,47,48,49] and enhances the subjects’ isolation from their environment. 

The results that achieve reliability at levels between good and excellent are the variables in the dynamic situation of the distance, the distance on x, and the distance on y both for SI and LI, and for OE and CE. This level of reliability may be due to the technical specifications of the platform (40 Hz), which improves the interactive algorithms for calculating the estimation of the values. The measurement points are double in static condition than in dynamic condition; it would reduce the calculation error in the approximations of the data in the different distances. Despite the fact that the ICCs with OE have levels of reliability between good and excellent, greater variability was still observed in the measurements with OE compared to CE for the different variables of distance. 

The protocol stipulates the conditions under which the measurements must be carried out, and which procedures must be followed. However, some factors are difficult to control, such as the fact that each subject exerts a slightly different pressure on the gauge which establishes the position of the subject’s feet on the platform, and this can lead to a noticeable difference in the initial position of the measurement, and consequently reduce the reliability [47].

As mentioned above, the reliability under static conditions was slightly lower than under dynamic conditions, which raises the question of whether it is necessary to increase the clearing period, especially under static conditions. In other words, this is the static and dynamic assessment, which when performed a second time in static it may be possible to improve the measurement, as opposed to the dynamic phase in which learning becomes more difficult due to the difficulty involved.

The study had limitations that need to be considered. The main limitation of this study was observed at phase II under static conditions. The subjects reported that the recording time was too long. This could have led to a decrease in reliability due to fatigue of the subjects.

As a projection of possible studies, patterns of stabilometric normality should be established in other age ranges and also in subjects with different pathologies. In this case it would be possible to correlate with the patterns of postural normality and determine how these pathologies can affect the postural control.

## 5. Conclusions

The Satel 40 Hz stabilometric force platform seems to be a valid and reliable tool for assessing both static and dynamic standing balance. It is valid for post-recording measurements by detecting changes in the position of the center of mass, and reducing the variability of the response as the weight to be assessed increases.

The Satel 40 Hz stabilometric force platform has good to excellent reliability for all values for stabilometric measurements for static and dynamic balance. It has excellent reliability for dynamic distance and sagittal imbalance with the eyes open, and dynamic distance by lateral imbalance with the eyes closed.

## Figures and Tables

**Figure 1 ijerph-17-07733-f001:**
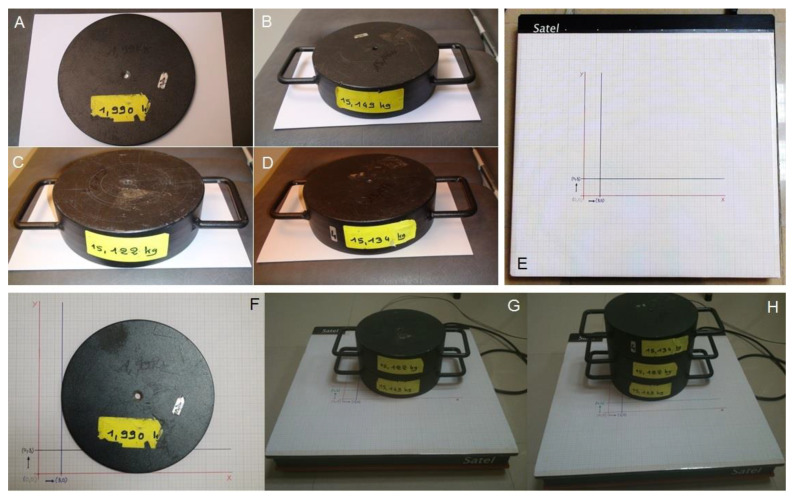
(**A**) Weight of 1.990 kg, (**B**) weight of 15.149 kg, (**C**) weight of 15.122 kg, (**D**) 15.134 kg, (**E**) Satel platform with graph paper, (**F**) weight position of 1.990 displaced on the x-axis, (**G**) weight position 30.271 shifted on the x-axis, (**H**) weight position 45.405 kg shifted on the y-axis.

**Figure 2 ijerph-17-07733-f002:**
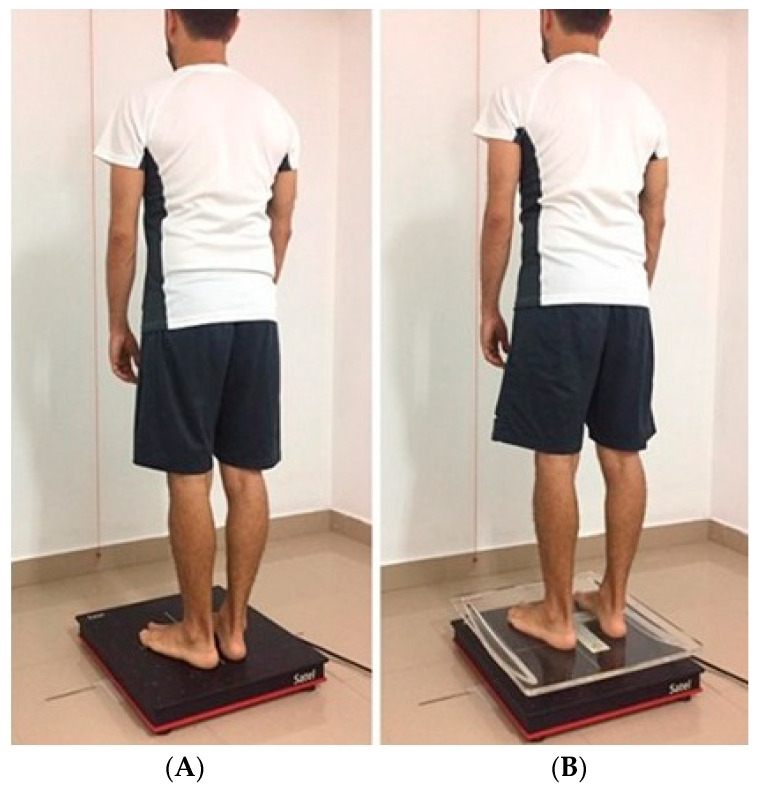
(**A**) Assessing quiet stance; (**B**) Assessing dynamic balance.

**Table 1 ijerph-17-07733-t001:** Results of force platform Satel validation by parameters Surface, X mean, Y mean, Length (L), Length on x-axis (LX), Length on y-axis (LY).

Parameters	Weight	(0,0)	(0,3)	(3,0)	% ∆	% ∆
	(kg)	(Mean ± SD)	(Mean ± SD)	(Mean ± SD)	*p*-Value (0,0)/(0,3)	*p*-Value (0,0)/(3,0)
Surface	1.990	6.14 ± 0.50	7.96 ± 0.59	5.28 ± 0.28	29.58%/0.000 *	−13.99%/0.001 *
	15.149	0.97 ± 0.10	1.10 ± 0.05	0.84 ± 0.03	13.20%/0.002 *	−13.00%/0.004 *
	30.271	0.79 ± 0.02	0.80 ± 0.06	0.68 ± 0.03	1.02%/0.661 *	−13.77%/0.000 *
	45.405	0.48 ± 0.03	0.87 ± 0.96	0.58 ± 0.03	80.26%/0.005 *^,w^	20.11%/0.000 *
X mean	1.990	−12.64 ± 0.12	−12.65 ± 0.13	18.32 ± 0.06	0.06%/0.721 *^,w^	−244.98%/0.000 *
	15.149	−8.93 ± 0.01	−9.08 ± 0.01	20.95 ± 0.01	1.77%/0.000 *	−334.73%/0.000 *
	30.271	−8.55 ± 0.00	−8.84 ± 0.00	21.57 ± 0.00	3.42%/0.005 *^,w^	−352.35%/0.005 *
	45.405	−8.39 ± 0.00	−8.75 ± 0.08	21.98 ± 0.01	4.31%/0.005 *^,w^	−362.02%/0.000 *
Y mean	1.990	103.58 ± 0.07	133.66 ± 0.19	103.92 ± 0.04	29.04%/0.005 *	0.33%/0.000 *
	15.149	108.05 ± 0.01	138.06 ± 0.01	107.75 ± 0.01	27.77%/0.000 *	−0.28%/0.000 *
	30.271	108.65 ± 0.00	138.21 ± 0.00	108.22 ± 0.00	27.21%/0.000 *	−0.40%/0.000 *
	45.405	108.84 ± 0.00	138.17 ± 0.03	108.41 ± 0.00	26.95%/0.005 * _w_	−0.39%/0.000 *
L	1.990	187.46 ± 2.67	193.01 ± 3.45	193.82 ± 2.09	2.96%/0.002 *	3.40% 7 0.001 *
	15.149	29.97 ± 1.94	28.34 ± 0.81	28.56 ± 0.40	−5.44%/0.028 *^,w^	−4.70%/0.047 *^,w^
	30.271	25.49 ± 0.38	22.72 ± 0.67	24.85 ± 0.59	−10.84%/0.000 *	−2.49%/0.017 *^,w^
	45.405	15.47 ± 0.45	14.88 ± 0.47	18.64 ± 0.41	−3.68%/0.037 *^,w^	20.66%/0.005 *^,w^
LX	1.990	117.42 ± 2.46	119.91 ± 2.37	124.62 ± 2.28	2.12%/0.037 *	6.13%/0.000 *
	15.149	18.86 ± 1.15	17.61 ± 0.55	18.27 ± 0.27	−6.61%/0.014 *	−3.11%/0.147
	30.271	13.41 ± 0.28	11.13 ± 0.40	13.75 ± 0.39	−17.01%/0.000 *	2.53%/0.121
	45.405	8.16 ± 0.26	7.62 ± 0.28	10.00 ± 0.32	−6.53%/0.001 *	22.57%/0.000 *
LY	1.990	129.39 ± 2.16	134.03 ± 3.33	129.41 ± 1.40	3.59%/0.006 *	0.02%/0.978
	15.149	20.10 ± 1.48	19.20 ± 0.58	18.61 ± 0.46	−4.52%/0.114 ^w^	−7.43%/0.013 *^,w^
	30.271	19.75 ± 0.40	18.09 ± 0.61	18.46 ± 0.41	−8.42%/0.005 *^,w^	−6.50%/0.005 *^,w^
	45.405	11.78 ± 0.42	11.48 ± 0.39	13.78 ± 0.39	−2.49%/0.193	17.00%/0.000 *

Note. Kg kilograms; SD standard deviation; * statistically significant; ^w^ Wilcoxon test.

**Table 2 ijerph-17-07733-t002:** Results of force platform Satel validation in amplitude, in the maximum and minimum values on axis x and axis y.

Parameters	Weight (kg)	(0,0)	(0,3)	(3,0)	% ∆	% ∆
		(Mean ± SD	(Mean ± SD)	(Mean ± SD)	*p*-Value (0,0)/(0,3)	*p*-Value (0,0)/(3,0)
Stabilometry Amplitude in X	1.990	14.74 ± 0.01	17.92 ± 0.02	14.78 ± 0.01	21.54%/0.005 *	0.26%/0.005 *
	15.149	15.22 ± 0.00	18.37 ± 0.00	15.19 ± 0.00	20.70%/0.002 *	−0.20%/0.002 *
	30.271	15.29 ± 0.00	18.39 ± 0.00	15.24 ± 0.00	20.27%/0.002 *	−0.33%/0.002 *
	45.405	15.31 ± 0.00	18.38 ± 0.01	15.26 ± 0.00	20.08%/0.004 *	−0.33%/0.002 *
Stabilometry in X maximum	1.990	−12.04 ± 0.21	−11.93 ± 0.22	18.96 ± 0.20	−0.86%/0.303	−257.53%/0.000 *
	15.149	−8.81 ± 0.03	−8.99 ± 0.03	21.07 ± 0.03	2.04%/0.000 *	−339.22%/0.000 *
	30.271	−8.46 ± 0.02	−8.76 ± 0.05	21.65 ± 0.03	3.64%/0.000 *	−356.05%/0.000 *
	45.405	−8.34 ± 0.01	−8.68 ± 0.08	22.06 ± 0.02	4.17%/0.000 *	−364.66%/0.000 *
Stabilometry in X minimum	1.990	−13.34 ± 0.23	−13.32 ± 0.23	17.76 ± 0.06	−0.19%/0.836	−233.11%/0.005 *^,w^
	15.149	−9.04 ± 0.06	−9.19± 0.02	20.86 ± 0.03	1.58%/0.000 *	−330.64%/0.000*
	30.271	−8.64 ± 0.02	−8.91 ± 0.02	21.48 ± 0.02	3.14%/0.000 *	−348.75%/0.000 *
	45.405	−8.45 ± 0.02	−8.81 ± 0.08	21.90 ± 0.04	4.23%/0.000 *	−358.99%/0.000 *
Stabilometry Amplitude in Y	1.990	−1.45 ± 0.01	−1.45 ± 0.02	2.10 ± 0.01	0.14%/0.587	−335.29%/0.005 *
	15.149	−1.02 ± 0.00	−1.04 ± 0.00	2.40 ± 0.00	1.96%/0.002 *	−244.89%/0.002 *
	30.271	−0.98 ± 0.00	−1.01 ± 0.00	2.47 ± 0.00	3.06%/0.002 *	−352.04%/0.002 *
	45.405	−0.96 ± 0.00	−1.00 ± 0.01	2.52 ± 0.00	4.48%/0.004 *	−362.50%/0.002 *
Stabilometry in Y maximum	1.990	102.80 ± 0.17	132.90 ± 0.27	103.21 ± 0.08	29.28%/0.000 *	0.40%/0.000 *
	15.149	107.93 ± 0.05	137.94 ± 0.02	107.66 ± 0.01	27.81%/0.000 *	−0.25%/0.000 *
	30.271	108.52 ± 0.02	138.10 ± 0.03	108.10 ± 0.02	27.26%/0.000 *	−0.39%/0.000 *
	45.405	108.74 ± 0.06	138.02 ± 0.08	108.27 ± 0.11	26.93%/0.000 *	−0.43%/0.005 *^,w^
Stabilometry in Y minimum	1.990	104.25 ± 0.14	134.39 ± 0.23	104.63 ± 0.11	28.91%/0.000 *	0.37%/0.000 *
	15.149	108.17 ± 0.02	138.17 ± 0.02	107.88 ± 0.04	27.74%/0.000 *	−0.27%/0.000 *
	30.271	108.81 ± 0.04	138.34 ± 0.06	108.36 ± 0.02	27.14%/0.000 *	−0.41%/0.000 *
	45.405	108.94 ± 0.06	138.26 ± 0.04	108.51 ± 0.10	26.92%/0.000 *	−0.39%/0.005 *^,w^

Note. * Statistically significant; kg: kilogram; ^w^ Wilcoxon; % ∆ increase percentage.

**Table 3 ijerph-17-07733-t003:** Stabilometric results of stabilometric variables in static condition.

Parameters	Visual Condition	R1 (Mean ± SD)	R2 (Mean ± SD)	Correlation *p*-Value	ICC [95% CI]
Surface	OE	172.75 ± 92.41	210.70 ± 143.09	0.529 (0.000 *)	0.651 [0.350, 0.812]
	CE	239.64 ± 137.03	202.01 ± 90.05	0.664 (0.000 *)	0.750 [0.549, 0.870]
X mean	OE	−1.21 ± 5.36	−0.87 ± 5.14	0.371 (0.016 *)	0.540 [0.145, 0.753]
	CE	0.02 ± 6.17	−0.40 ± 5.36	0.635 (0.000 *)	0.772 [0.577, 0.878]
Y mean	OE	−34.10 ± 13.17	−36.60 ± 13.86	0.776 (0.000 *)	0.874 [0.765, 0.932]
	CE	−31.13 ± 12.41	−33.02 ± 12.24	0.761 (0.000 *)	0.864 [0.747, 0.927]
L	OE	463.11 ± 106.16	458.02 ± 104.05	0.545 (0.000 *)	0.705 [0.452, 0.842]
	CE	649.91 ± 209.16	579.22± 162.25	0.612 (0.000 *)	0.744 [0.524, 0.862]
LX	OE	266.23 ± 65.43	271.13 ± 69.21	0.599 (0.000 *)	0.749 [0.532, 0.865]
	CE	362.89 ± 117.10	326.73 ± 112.85	0.694 (0.000 *)	0.819 [0.664, 0.903]
LY	OE	321.65 ± 82.23	309.54 ± 77.72	0.477 (0.001 *)	0.645 [0.340, 0.809]
	CE	458.96 ± 169.92	407.26 ± 115.34	0.549 (0.000 *)	0.675 [0.396, 0.825]
Stabilometry Amplitude in X	OE	16.05 ± 5.97	16.25 ± 5.60	0.598 (0.000 *)	0.747 [0.530, 0.864]
	CE	21.49 ± 8.67	19.46 ± 5.89	0.635 (0.000 *)	0.742 [0.520, 0.861]
Stabilometry in X maximum	OE	7.13 ± 6.89	7.26 ± 6.09	0.407 (0.007 *)	0.575 [0.210, 0.772]
	CE	10.66 ± 8.51	8.99 ± 6.22	0.597 (0.000 *)	0.725 [0.488, 0.852]
Stabilometry in X minimum	OE	−8.91 ± 5.47	−8.99 ± 6.32	0.402 (0.008 *)	0.570 [0.199, 0.769]
	CE	−10.82 ± 7.39	−10.47 ± 6.08	0.523 (0.000 *)	0.679 [0.402, 0.827]
Stabilometry Amplitude in Y	OE	21.57 ± 5.76	24.36 ± 9.69	0.303 (0.051)	0.420 [−0.077, 0.689]
	CE	24.09 ± 7.35	22.29 ± 5.62	0.645 (0.000 *)	0.767 [0.567, 0.875]
Stabilometry in Y maximum	OE	−44.96 ± 13.70	−48.87 ± 16.48	0.657 (0.000 *)	0.785 [0.600, 0.885]
	CE	−42.99 ± 13.01	−43.98 ± 12.69	0.745 (0.000 *)	0.854 [0.728, 0.922]
Stabilometry in Y minimum	OE	−23.39 ± 13.50	−24.51 ± 14.31	0.765 (0.000 *)	0.866 [0.751, 0.928]
	CE	−18.90 ± 13.73	−21.68 ± 13.12	0.743 (0.000 *)	0.852 [0.724, 0.920]

Note. CE close eyes; ICC: intra-class correlation coefficient; CI: confidence interval; OE open eyes; R repetition; * statistically significant.

**Table 4 ijerph-17-07733-t004:** Stabilometric results of stabilometric parameters in dynamic condition. Anterior–Posterior and Medial–Lateral Imbalance.

Parameters	Imbalance	Visual Condition	R1 (Mean ± SD)	R2 (Mean ± SD)	Correlation *p*-Value	ICC [95% CI]
Surface	SI	OE	408.97 ± 133.83	432.81 ± 168.34	0.525 (0.000 *)	0.677 [0.399, 0.826]
		CE	1331.15 ± 663.81	1213.30 ± 588.70	0.713 (0.000 *)	0.829 [0.682, 0.908]
	LI	OE	381.82 ± 165.02	397.91± 149.48	0.400 (0.009 *)	0.570 [0.200, 0.769]
		CE	1669.57 ± 800.46	1607.78 ± 720.49	0.629 (0.000 *)	0.769 [0.571, 0.876]
X mean	SI	OE	−1.90 ± 7.80	0.48 ± 6.37	0.480 (0.001 *)	0.640 [0.330, 0.806]
		CE	−0.58 ± 7.42	0.20 ± 7.59	0.480 (0.001 *)	0.648 [0.346, 0.811]
	LI	OE	−2.19 ± 6.32	−1.88 ± 6.94	0.356 (0.021 *)	0.523 [0.113, 0.744]
		CE	−4.29 ± 9.97	−4.51 ± 7.13	0.587 (0.000 *)	0.714 [0.468, 0.846]
Y mean	SI	OE	−2.93 ± 18.10	−5.40± 14.39	0.575 (0.000 *)	0.718 [0.475, 0.848]
		CE	−1.98 ± 20.66	−8.62 ± 17.31	0.788 (0.000 *)	0.874 [0.766, 0.932]
	LI	OE	−3.51 ± 15.83	−7.9127 ± 17.38	0.743 (0.000 *)	0.851 [0.722, 0.920]
		CE	3.62 ± 16.98	−1.5315 ± 16.89	0.765 (0.000 *)	0.867 [0.752, 0.928]
L	SI	OE	703.91 ± 189.26	645.94 ± 176.18	0.823 (0.000 *)	0.901 [0.816, 0.947]
		CE	1346.19 ± 543.73	1220.79 ± 420.82	0.812 (0.000 *)	0.880 [0.778, 0.936]
	LI	OE	594.75 ± 192.48	565.51 ± 142.08	0.795 (0.000 *)	0.864 [0.746, 0.927]
		CE	1230.91 ± 404.83	1160.31 ± 384.42	0.829 (0.000 *)	0.906 [0.824, 0.949]
LX	SI	OE	247.42± 71.16	226.36 ± 75.15	0.628 (0.000 *)	0.771 [0.573, 0.877]
		CE	491.79 ± 178.74	472.93 ± 212.37	0.762 (0.000 *)	0.858 [0.735, 0.923]
	LI	OE	427.17 ± 159.23	409.11 ± 123.16	0.808 (0.000 *)	0.878 [0.773, 0.934]
		CE	882.52 ± 292.44	835.60 ± 275.34	0.808 (0.000 *)	0.893 [0.801, 0.942]
LY	SI	OE	603.37 ± 182.31	555.24 ± 161.00	0.822 (0.000 *)	0.898 [0.811, 0.945]
		CE	1143.29 ± 499.99	1017.34 ± 354.42	0.773 (0.000 *)	0.844 [0.709, 0.916]
	LI	OE	319.60 ± 96.75	299.73 ± 74.62	0.703 (0.000 *)	0.809 [0.645, 0.897]
		CE	664.13 ± 245.79	620.75 ± 233.48	0.801 (0.000 *)	0.889 [0.793, 0.940]
Starbilometry Amplitude in X	SI	OE	18.87 ± 5.57	19.02 ± 5.34	0.538 (0.000 *)	0.699 [0.440, 0.838]
		CE	34.65 ± 10.58	33.10 ± 12.18	0.598 (0.000 *)	0.744 [0.524, 0.862]
	LI	OE	29.95 ± 10.46	28.87 ± 7.28	0.474 (0.002 *)	0.615 [0.284, 0.793]
		CE	72.48 ± 21.28	71.98 ± 18.93	0.469 (0.002 *)	0.636 [0.323, 0.804]
Stabilometry in X maximum	SI	OE	7.75 ± 9.41	9.90 ± 6.92	0.466 (0.002 *)	0.616 [0.286, 0.794]
		CE	16.33 ± 9.57	16.77 ± 10.53	0.538 (0.000 *)	0.697 [0.437, 0.837]
	LI	OE	12.72 ± 8.49	12.47 ± 8.29	0.246 (0.116)	0.395 [−0.013, 0.675]
		CE	31.68 ± 16.07	31.78 ± 14.45	0.513 (0.001 *)	0.676 [0.397, 0.826]
Stabilometry in X minimum	SI	OE	−11.11 ± 8.36	−9.11 ± 7.23	0.431 (0.004 *)	0.613 [0.280, 0.792]
		CE	−18.32 ± 8.99	−16.32 ± 8.93	0.532 (0.000 *)	0.694 [0.432, 0.836]
	LI	OE	−17.22 ± 9.59	−16.40 ± 7.24	0.402 (0.008 *)	0.558 [0.177, 0.762]
		CE	−40.79 ± 14.12	−40.19 ± 12.03	0.509 (0.001 *)	0.669 [0.384, 0.822]
Stabilometry Amplitude in Y	SI	OE	41.57 ± 10.09	42.83 ± 9.39	0.360 (0.019 *)	0.529 [0.123, 0.747]
		CE	74.12 ± 18.52	71.68 ± 18.56	0.630 (0.000 *)	0.773 [0.577, 0.878]
	LI	OE	25.46 ± 6.99	25.68 ± 8.28	0.433 (0.004 *)	0.598 [0.252, 0.784]
		CE	44.40 ± 14.65	41.17 ± 14.05	0.553 (0.000 *)	0.712 [0.464, 0.845]
Stabilometry in Y maximum	SI	OE	−23.868 ± 16.35	−26.75 ± 13.88	0.499 (0.001 *)	0.660 [0.386, 0.817]
		CE	−39.21 ± 20.90	−43.43 ± 17.77	0.764 (0.000 *)	0.860 [0.739, 0.925]
	LI	OE	−16.21 ± 15.65	−20.78 ± 16.94	0.687 (0.000 *)	0.813 [0.652, 0.900]
		CE	−19.05 ± 17.41	−22.29 ± 17.53	0.731 (0.000 *)	0.845 [0.711, 0.916]
Stabilometry in Y minimum	API	OE	17.70 ± 20.58	16.07 ± 17.06	0.582 (0.000 *)	0.727 [0.493, 0.853]
		CE	34.90 ± 25.68	28.25 ± 21.91	0.656 (0.000 *)	0.786 [0.603, 0.885]
	LI	OE	9.24 ± 15.86	4.90 ± 18.33	0.765 (0.000 *)	0.862 [0.740, 0.926]
		CE	25.35 ± 20.63	18.88 ± 19.25	0.729 (0.000 *)	0.842 [0.706, 0.915]

Note. API: antero-posterior imbalance; CE: close eyes; CI: confidence interval; LI: lateral imbalance; OE: open eyes; SI: sagittal imbalance; * statistically significant.

## Data Availability

Validity and reliability of the satel 40 hz stabilometric force platform for measuring quiet stance and dynamic standing balance in healthy subjects (view at https://dataverse.harvard.edu/dataset.xhtml?persistentId=doi:10.7910/dvn/mpbjnu) was published in Harvard Dataverse (https://dataverse.harvard.edu/dataset.xhtml?persistentId=doi:10.7910/DVN/MPBJNU).

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
