# Peer review of "Validity and Reliability of the Satel 40 Hz Stabilometric Force Platform for Measuring Quiet Stance and Dynamic Standing Balance in Healthy Subjects"

_ijerph, 2020, doi:10.3390/ijerph17217733_

Round 1

Reviewer 1 Report

Congratulations on this study.

After reviewing and analyzing the manuscript, I proceed to describe the changes that can be made and explain the appropriate comments to improve it.

Abstract
In this section you have to add more objective results, which show the validity and reliability of the described platform.

Introduction
In line 31, you affirm that the force platform is the Gold Standard for the assessment of postural control, but are all the pressure platforms Gold Standard or only some are?
You should look for more references that justify this study, why is it important to validate this tool for health professionals, researchers or patients?

Methods
You must describe the specific characteristics of the force platform and then you must describe phases I and II of the study.
Phase I: Why are the measurements to evaluate the validity of the platform repeated 10 times and not repeated anymore?
Phase II: Why is the age range of the selected subjects 18 to 25 and not of older subjects? You must justify this inclusion criterion. Was an age range assessment performed for a specific reason?
What is the purpose of the plumb line? explain it.
You must justify and explain why the exclusion criteria of this study were applied.

Results
Good results obtained.
The results have been correctly described and are easy to interpret.

Discussion
The characteristics of the position and the specific requirements that patients had to meet were discussed.
Do you think it would have been better to use a fixed point on the wall (in front of the patient) instead of using the plumb line? With the aim that the patient with open eyes had a more specific point to look at.

The limitations of this study should be better specified in this discussion section.
It is also convenient to describe the possible studies that can be made derived from it.

Why hasn't another tool like Gold Standard been used to perform a better validation of this force platform? In order to make a correlation between the two tools in Phase II of this study.
For the latter, it seems very conclusive to affirm that the reliability of this pressure platform is excellent without having compared it with another tool.

Have you obtained any funding?

I look forward to seeing the changes made to this manuscript. It is a good study and the correct validation of this type of tool is very important.

Thank
you

Author Response

Abstract
In this section, you have to add more objective results, which show the validity and reliability of the described platform.

The results section of the Abstract has been modified following the indications.

The validity increased as the weight increased for all the variables measured in the stabilometric parameters (p<0.05). The reliability was shown to be good to excellent for the two visual conditions. The positional variables obtained a higher ICC. The variable with the best ICC was the Y mean in OE (ICC 0.874 and a p<0.001). All the values showed an increase in a dynamic situation.

Introduction
In line 31, you affirm that the force platform is the Gold Standard for the assessment of postural control, but are all the pressure platforms Gold Standard or only some are?

This sentence has been modified by lowering the expression and two references were added.

According to the specialized literature, the force platform is effectively the most used instrument to evaluate postural stability.

You should look for more references that justify this study, why is it important to validate this tool for health professionals, researchers or patients?

There are studies that used this platform; however, there is no specific validation that supports its use.

The number of references justifying its use has increased.

Methods
You must describe the specific characteristics of the force platform and then you must describe phases I and II of the study.

Both the platform and phases I and II of the study have been described more in detail.

The validity of the Satel 40 Hz stabilometric force platform (model PF2002; SATEL SARL, 6 rue du Limousin – 31700 Blagnac; France) was calculated using the v. 33.5 8C software. The voltage is 230V, 50/60Hz, 2A with USB cable output. Classified as a class I-type B device. It uses three sensors with a sensitivity per sensor of 2.0 mV/V±0.1.

In phase I the instrument is validated to determine the change in the position of constant weight on the platform.

In phase II, the purpose is to know the reliability of the measurements on people where there is an oscillation in their centre of mass.

Phase I: Why are the measurements to evaluate the validity of the platform repeated 10 times and not repeated anymore?

Ten repetitions were used by internal consensus. As this was a constant weight, ten repetitions were considered sufficient to obtain accurate data for each repetition.

Phase II: Why is the age range of the selected subjects 18 to 25 and not of older subjects? You must justify this inclusion criterion. Was an age range assessment performed for a specific reason?

The age range was established in order to obtain a sample with fewer pathologies that could interfere with the postural control.

What is the purpose of the plumb line? explain it.

The following sentence has been added to the text.  

A plumb line is a weight suspended from a string used as a vertical reference line to ensure a structure is centered. As the plum line always finds the vertical axis pointing to the center of gravity, it ensures everything is right, justified, and centered.

You must justify and explain why the exclusion criteria of this study were applied.

Subjects that may present symptoms or pathologies which may affect the postural control such as pain, musculoskeletal or neurological pathologies, use of plantar templates that make the subject in the process of adaptation, etc. were discarded.

Results
Good results obtained.

Thank you very much.

The results have been correctly described and are easy to interpret.

Thank you very much.

Discussion
The characteristics of the position and the specific requirements that patients had to meet were discussed.
Do you think it would have been better to use a fixed point on the wall (in front of the patient) instead of using the plumb line? With the aim that the patient with open eyes had a more specific point to look at.

The plummet allows the subject to adjust his vision to a point on the line. One point is conditioned to the height of the patient and this will modify the cervical posture. We do not know how this could have affected the results.

The limitations of this study should be better specified in this discussion section.
It is also convenient to describe the possible studies that can be made derived from it.
The main limitation of this study was observed in phase II under static conditions. The subjects reported that the recording time was too long. This could have led to a decrease in reliability due to fatigue of the subjects.

A projection of possible studies to carry out is to establish patterns of stabilometric normality in other age ranges and also in patients presenting different pathologies. In this way, it would be possible to compare the new data with the patterns of postural normality and determine if these pathologies affect the postural control.

A limitation paragraph has been created to include this explanation.

The study had limitations that need to be considered. The main limitation of this study was observed at phase II under static conditions. The subjects reported that the recording time was too long. This could have led to a decrease in reliability due to the subject’s fatigue.

As a projection of possible studies, patterns of stabilometric normality should be established in other age ranges and also in subjects with different pathologies. In this case it would be possible to correlate with the patterns of postural normality and determine how these pathologies can affect the postural control.

Why hasn't another tool like Gold Standard been used to perform a better validation of this force platform? In order to make a correlation between the two tools in Phase II of this study.

For the latter, it seems very conclusive to affirm that the reliability of this pressure platform is excellent without having compared it with another tool.

There are several reasons. The main reason is that we did not have any other validated platform. Another reason is the lack of consensus on the variables obtained by the different manufacturers, some of which may be referred to as surface area and length, but not to other variables provided in this study. The variables presented in this study describe how to quantify the postural control of the subjects in a better way.

Have you obtained any funding?

No, we did not obtain any funding.

I look forward to seeing the changes made to this manuscript. It is a good study and the correct validation of this type of tool is very important.

Thank you very much for your appreciation

Reviewer 2 Report

The content of the article and the theme is appropriate, but for its publication it would be necessary to make some modifications. Pictures don't add much. The tables are too long, and table 5 is at the end. I would reduce the content of the tabals. It is necessary to provide the most recent bibliography.

Author Response

The content of the article and the theme is appropriate, but for its publication, it would be necessary to make some modifications. Pictures don't add much. The tables are too long, and table 5 is at the end. I would reduce the content of the tables. It is necessary to provide the most recent bibliography.

Thank you very much for your appreciation

We believe that figures and tables are necessary for a better understanding of the process and the results. If the editor feels that it is necessary to remove them or to move part of them to supplementary material, this would not be a problem for authors.

However, we have removed table 1, which is already cited in the text.

The tables have been renamed again, as an error was detected in the numbering.

Reviewer 3 Report

This study primarily examines the reliability and validity of the Satel 40 Hz force platform. The findings reported by the authors are promising in that sense. Based on their observations, the authors discussed that the platform is reliable to evaluate static and dynamic postural stability in healthy adult individuals. It could further be argued that the platform could also be used reliably with children considering its sensitivity to displacement of loads similar to the weight of children.

Specific comments:

Line 84: The description of the figures 1C and D should be accurate; they are not showing the weights 30.271 Kg and 45.405 Kg.

Line 128: Please clarify what was the low intensity activity. Was this likely to induce some form of fatigue, and how was this ruled out?

Line 132: It is not clear how the imbalances were produced by the methacrylate plate. What was the nature and frequency of the imbalances? Were they translations in the anteroposterior and mediolateral planes?

Line 134: Did the participants show any sign of destabilisation in the anteroposterior and lateral dynamic conditions with OE that could have been exacerbated with CE?

Line 143: The authors should be cautious when interpreting this result. The Spearman's correlation determines the strength and direction of a monotonic relationship between the two variables, but not the strength and direction of a linear relationship between the two variables, which is determined by a Pearson's correlation.

Line 239: One can disagree with the claim that changes in the position of centre of pressure were measured since there is no report of measures of vertical force variations. Here the authors are reporting movement of the different loads from one point to another.

Line 251: Could the authors please explain whether the platform needs to be re-calibrated (zeroed) before moving the loads, which might explain this observation, and also between conditions.

Line 293: The mental calculation argued here assumes a dual tasking process, which cannot be accounted for if not explicitly measured. Hence, it is not clear how this discussion is relevant.

Author Response

This study primarily examines the reliability and validity of the Satel 40 Hz force platform. The findings reported by the authors are promising in that sense. Based on their observations, the authors discussed that the platform is reliable to evaluate static and dynamic postural stability in healthy adult individuals. It could further be argued that the platform could also be used reliably with children considering its sensitivity to displacement of loads similar to the weight of children.

Thank you for your comments.

Records could be made on subjects weighing more than 15 kg, and this should be studied. But the time required to record the test in position, makes its use in children more difficult, introducing confusing parameters.

Specific comments:

Line 84: The description of the figures 1C and D should be accurate; they are not showing the weights 30.271 Kg and 45.405 Kg.

Figures 1C and 1D are calibrated weights that, added to the weights represented in images 1A and 1B, make up the weights of 1G and 1D. The 4 weights used in phase I of the study that was calibrated are shown.

Line 128: Please clarify what was the low-intensity activity. Was this likely to induce some form of fatigue, and how was this ruled out?

The low-intensity activity was to walk around the room in a calm and leisurely manner.

We do not consider this to be a great fatigue, but it has been added in a paragraph that mentions this possible limitation.

The main limitation of this study was observed at phase II under static conditions. Subjects reported that the recording time was too long. This could have led to a decrease in reliability due to subject’s fatigue.

Line 132: It is not clear how the imbalances were produced by the methacrylate plate. What was the nature and frequency of the imbalances? Were they translations in the anteroposterior and mediolateral planes?

The platform has two crescents in its lower part that generate an anterior-posterior unbalance, as can be seen in figure 2. This unbalance can be lateral by changing the position of the methacrylate plate.

Line 134: Did the participants show any sign of destabilisation in the anteroposterior and lateral dynamic conditions with OE that could have been exacerbated with CE?

Imbalance with closed eyes adds a further difficulty in maintaining balance, especially in the initial moments.

A worsening was not observed when making the recording in CE as it can be seen in the Stabilometry Amplitude in Y variable where in CE the average of ICC is 0.767, while in OE the ICC is 0.420.

Line 143: The authors should be cautious when interpreting this result. The Spearman's correlation determines the strength and direction of a monotonic relationship between the two variables, but not the strength and direction of a linear relationship between the two variables, which is determined by a Pearson's correlation.

The text has been modified following the reviewer's suggestions.

The Pearson or Spearman correlation coefficient was calculated to determine the force and direction between the variable measurements.

Line 239: One can disagree with the claim that changes in the position of the center of pressure were measured since there is no report of measures of vertical force variations. Here the authors are reporting the movement of the different loads from one point to another.

We understand that the center of pressure to be the intermediate point of the support base formed by both feet and calculated by the length of the foot. This makes it possible to determine the movement of the center of pressures and which is picked up by the platform's sensors. If it is true, that the vertical movements that this center of pressures can go through have not been able to be analyzed due to the limitations of the platform.

Line 251: Could the authors please explain whether the platform needs to be re-calibrated (zeroed) before moving the loads, which might explain this observation, and also between conditions.

The calibration is done at the beginning, once the machine is switched on. If during use any sensor loses the signal, it must be recalibrated as a requirement by the platform's own software.

Line 293: The mental calculation argued here assumes a dual-tasking process, which cannot be accounted for if not explicitly measured. Hence, it is not clear how this discussion is relevant.

Some authors consider them as a factor that adds difficulties to the strategy of maintaining the balance. In the present study, the fact that the subject performed cognitive tasks during the recording was not taken into account.